Experimental validation of otolith-based age and growth reconstructions across multiple life stages of a critically endangered estuarine fish

Xieu Wilson 1
Lewis Levi S. lewis.sci@gmail.com 1
Zhao Feng 1
Fichman Rachel A. 1
Willmes Malte 2 3
Hung Tien-Chieh 4
Ellison Luke 4
Stevenson Troy 4
Tigan Galen 4
Schultz Andrew A. 5
Hobbs James A. 1 6
1 Department of Wildlife, Fish, and Conservation Biology, University of California , Davis , CA , United States of America
2 Institute of Marine Sciences, University of California , Santa Cruz , CA , United States of America
3 Southwest Fisheries Science Center, National Marine Fisheries Service , Santa Cruz , CA , United States of America
4 Department of Biological and Agricultural Engineering, University of California , Davis , CA , United States of America
5 Bay-Delta Office, United States Bureau of Reclamation , Sacramento , CA , United States of America
6 Bay-Delta Region, California Department of Fish and Wildlife , Stockton , CA , United States of America
Posner Mason
Electronic publication date: 2021 Nov 17
Publication date: 2021
Volume: 9
Electronic Location ID: e12280
Received 2021 Feb 24; Accepted 2021 Sep 20
Copyright: ©2021 Xieu et al.
Copyright year: 2021
Copyright holder: Xieu et al.
License: This is an open access article distributed under the terms of the Creative Commons Attribution License, which permits unrestricted use, distribution, reproduction and adaptation in any medium and for any purpose provided that it is properly attributed. For attribution, the original author(s), title, publication source (PeerJ) and either DOI or URL of the article must be cited.
License URL: https://creativecommons.org/licenses/by/4.0/

Keywords: Otolith, Fish, Estuarine, Freshwater, Growth, Validation, Delta Smelt, San Francisco

Funding: The U.S. Bureau of Reclamation Directed Outflow Project (No. R17AC00129) This work was supported by the U.S. Bureau of Reclamation Directed Outflow Project (No. R17AC00129). A. Schultz (USBR) provided logistical support for carrying out the work and valuable feedback in the drafting of the final manuscript.

==============================
Background

The application of otolith-based tools to inform the management and conservation of fishes first requires taxon- and stage-specific validation. The Delta Smelt (Hypomesus transpacificus), a critically endangered estuarine fish that is endemic to the upper San Francisco Estuary (SFE), California, United States, serves as a key indicator species in the SFE; thus, understanding this species’ vital rates and population dynamics is valuable for assessing the overall health of the estuary. Otolith-based tools have been developed and applied across multiple life stages of Delta Smelt to reconstruct age structure, growth, phenology, and migration. However, key methodological assumptions have yet to be validated, thus limiting confidence in otolith-derived metrics that are important for informing major water management decisions in the SFE.

Methods

Using known-age cultured Delta Smelt and multiple independent otolith analysts, we examined otolith formation, otolith-somatic proportionality, aging accuracy and precision, left-right symmetry, and the effects of image magnification for larval, juvenile, and adult Delta Smelt.

Results

Overall, otolith size varied linearly with fish size (from 10–60 mm), explaining 99% of the variation in fish length, despite a unique slope for larvae < 10 mm. Otolith-somatic proportionality was similar among wild and cultured specimens. Aging precision among independent analysts was 98% and aging accuracy relative to known ages was 96%, with age estimates exhibiting negligible differences among left and right otoliths. Though error generally increased with age, percent error decreased from 0–30 days-post-hatch, with precision remaining relatively high (≥ 95%) thereafter. Increased magnification (400×) further improved aging accuracy for the oldest, slowest-growing individuals. Together, these results indicate that otolith-based techniques provide reliable age and growth reconstructions for larval, juvenile, and adult Delta Smelt. Such experimental assessments across multiple developmental stages are key steps toward assessing confidence in otolith-derived metrics that are often used to assess the dynamics of wild fish populations.

Introduction

The assessment and management of fish populations require knowledge regarding the age-structure, mortality, growth, phenology, and migratory history of each species (Maunder & Punt, 2013). Such information is particularly valuable for endangered species, where high stakes and high uncertainty can hinder the development of effective conservation policies (Meffe, 1986; Runge, 2011). The application of schlerochronology, the study of calcareous age-registering accretionary body parts such as otoliths, vertebrae, and fin spines, in fisheries science has provided several tools to help assess the status and dynamics of managed fish populations (Hunter, Laptikhovsky & Hollyman, 2018; Trofimova et al., 2020).

Otoliths (ear stones) are paired calcium carbonate structures found in the inner ears of bony fishes that are inert and accrete continuously throughout the life of a fish (Pannella, 1971; Campana, 1999). Otolith accretion often results in daily or annual ring patterns that can be used to quantify a fish’s age while also providing a permanently archived chronology of its growth and environmental history (Pannella, 1971; Campana & Neilson, 1985; Campana, 1999; Campana & Thorrold, 2001; Starrs, Ebner & Fulton, 2016). Otoliths, therefore, can be used to reconstruct the life history (Hobbs et al., 2010; Hobbs et al., 2019; Gillanders et al., 2015; Rogers et al., 2019) and vital rates (Feyrer, Sommer & Hobbs, 2007; Black et al., 2011; Martino et al., 2019) of fishes, thus improving our understanding of their population dynamics and movement patterns (Campana, 1999; Starrs, Ebner & Fulton, 2016; Willmes et al., 2018).

These data are critical for developing effective management plans for endangered species such as California’s Delta Smelt (Hypomesus transpacificus). The Delta Smelt is an estuarine osmerid smelt that is endemic to the San Francisco Estuary, California, United States. Delta Smelt generally exhibit an annual life cycle and a complex migratory life-history (Moyle et al., 1992; Hobbs et al., 2019). Though this forage fish was historically abundant throughout the upper SFE, the population has steeply declined since the 1980s, likely due to multiple factors including pollution, invasive species, habitat loss, hydrologic modifications, and changing environmental conditions (Feyrer, Nobriga & Sommer, 2007; Sommer et al., 2007; Moyle et al., 2016; Hobbs et al., 2017; Moyle, Hobbs & Durand, 2018). As a result, Delta Smelt are listed as threatened, endangered, and critically endangered under the federal Endangered Species Act (ESA), the California Endangered Species Act (CESA), and the International Union for Conservation of Nature (IUCN) Red List, respectively (U.S. Fish and Wildlife Service, 1993; CDFG, 2010; NatureServe, 2014).

The conservation status of Delta Smelt has resulted in several efforts to protect the species, including setting limits on freshwater exports that directly and indirectly impact the Delta Smelt population through entrainment and habitat modification (Grimaldo et al., 2009; Sommer et al., 2011; Miller et al., 2012; Moyle, Hobbs & Durand, 2018; Hammock et al., 2019; Smith, Newman & Mitchell, 2020). These restrictions on water exports have placed Delta Smelt in the crossfire between conserving species and providing a stable water supply to California’s 25 million southern residents and multi-billion dollar agriculture industry (Moyle, Hobbs & Durand, 2018). As a result, studies addressing the habitat needs and responses of Delta Smelt to natural and anthropogenic perturbations have become a key priority for managers and researchers in the region (Hobbs et al., 2017). Key elements of this include quantifying the age structure, hatch dates, movement patterns, and growth rates of Delta Smelt, all of which can potentially be obtained via otolith analysis.

Before otoliths can be used to inform the management of fish populations, several aspects of their preparation and interpretation must first be assessed experimentally including increment periodicity (accuracy), inter-operator error (precision), and consistency in otolith-somatic size relationships (proportionality) (Campana, 2001; Campana & Thorrold, 2001). To accomplish this, otolith validation studies are often conducted utilizing known-age cultured fish or marked-recaptured wild fish (Miller & Storck, 1982; Hoff, Logan & Markle, 1997; Campana, 2001; Roberts et al., 2004; Black, Boehlert & Yoklavich, 2005; Hobbs et al., 2007; Sakaris & Irwin, 2008; Sakaris, Buckmeier & Smith, 2014; Buckmeier & Howells, 2003). The application of otolith-based approaches to fisheries management requires that such validations are conducted across multiple developmental stages for a given species; thus, multi-stage experiments remain a key step toward improving confidence and value of otolith-derived metrics.

Objectives

Otolith-based tools have been applied across multiple life stages of several imperiled osmerid smelts in the SFE (e.g., Delta Smelt and Longfin Smelt) to inform conservation and management actions (Hobbs et al., 2010; Hobbs et al., 2019; Lewis et al., 2021). Key assumptions of these methods, however, have yet to be validated for all relevant age classes, thus limiting confidence in otolith-derived metrics that are important for informing major conservation and water management decisions in the system. Experimental validation of otolith tools for these threatened and endangered species has been identified as a critical need by state and federal resource management agencies (e.g., California Department of Fish and Wildlife-CDFW and the United States Bureau of Reclamation-USBR). Here we used known-age Delta Smelt that were cultured at the UC Davis Fish Conservation and Culture Laboratory (FCCL) to examine microstructure periodicity (aging accuracy), inter-operator error (aging precision), and consistency in the otolith-somatic size relationship (proportionality). Importantly, we aimed to expand the validated age range by 300%, including larvae, juveniles, and young adults; and to examine how key methodological considerations affect accuracy and repeatability of otolith-based fisheries techniques. Specifically, we examined fish age, otolith-somatic proportionality, left–right symmetry, initial increment formation, the accuracy, precision, and bias of age estimates, and the effects of image magnification for older, slower growing individuals. Furthermore, we contrasted otolith-somatic proportionality among cultured and wild Delta Smelt populations to assess the application of our results to the wild population. Results of this study are a valuable step toward improving confidence in past and future otolith-based age, growth, and geochemical studies which are key for informing population models and policy and management decisions.

Materials and Methods

Rearing and collection of specimens

Laboratory-reared (F11) mature Delta Smelt were spawned, and the larvae reared in 2018-2019 at the UC Davis Fish Conservation and Culture Laboratory (FCCL) following standard methods approved by the UC Davis Institutional Animal Care and Use Committee Protocol No. 19747 (Lindberg et al., 2013). In short, fertilized eggs were incubated in columns until hatch, and all fish were held in fresh water at 16 °C. For feed, larvae (<80 days-post-hatch, dph) received rotifers and Artemia sp. nauplii, juveniles (80–120 dph) received Artemia sp. Nauplii and Bio-Oregon BioVita Starter Mash (pellet food), and older juveniles and adults (>120 dph) received Bio-Oregon BioPro2 Crum#1, each provided ad libitum. During culture, tanks were checked daily, and fish that were either exhibiting signs of stress or collected for archival were euthanized in 500 mg/L MS-222. Larvae were archived in 95% ethanol at 0, 5, 10, 30, 61, and 90 dph, and adults were archived in a −20 °C freezer at 180, 215, 243, and 271 dph (20 per time point). Approximately 10 archived fish were selected for each time point to examine otolith development (Table 1). Sex was not included in the analysis because all fish were less than 1 year of age, and Delta Smelt do not exhibit sexual dimorphism (Wang, 2007). To contrast results with wild fish, otolith-somatic proportionality was assessed for 117 wild Delta Smelt collected and archived by the 2019 US Fish and Wildlife Service’s Delta Juvenile Fish Monitoring Program (EDSM Kodiak survey, https://www.fws.gov/lodi/juvenile_fish_monitoring_program/).

Length measurements

Adult Delta Smelt (≥ 180 dph) were imaged with a mounted Canon Powershot digital camera (Canon Solutions America Inc., Melville, New York, USA), and larvae (≤ 90 dph) were imaged at 20x magnification with an AmScope MU1000 10MP camera (AmScope, Irvine, California, USA) on a Leica StereoZoom7 dissecting microscope (Leica Camera Inc., Allendale, New Jersey, USA). All images included millimeter markers to facilitate image calibration and measurements. Digital measurements of standard length (SL), fork length (FL), and total length (TL) (Fig. 1A) were collected for each fish using ImageJ (version 1.8.0) (Abramoff, Magelhaes & Ram, 2004). Digital and hand measurements of standard length yielded nearly identical measurements (mean difference = 0.57 mm or 1.08%) (Fig. S1), thus were treated interchangeably as needed. To correct for preservation effects (Fowler & Smith, 1983; Fey, 1999), fresh “corrected” standard lengths (SLc) were calculated using empirical linear models developed in the laboratory for specimens that were measured fresh (SLf or TLf) and then preserved in 95% ethanol (SLe, slope = 1.018, intercept = 0.952, R2 = 0.977, n = 35) or were frozen at −20 °C (SLz, slope = 1.018, intercept = 1.524, R2 = 0.980, n = 36) (Fig. S1). The smallest larvae in the otolith study, however, were beyond the size range used to develop the ethanol conversion function, resulting in over-estimation of larval fish sizes. Therefore, the mean proportional difference was applied in lieu of linear correction functions (Fig. S1).

Table 1 Samples used in the present study.

Sample size (N) is shown by age class, along with the mean standard length for the subsample from each age class. The number of specimens for which paired left and right otoliths were used to examine symmetry (Sym) and paired images at 200× and 400× magnification to examine magnification (Mag) are also provided.

Age (dph)	N (fish)	SLc (mm)	Sym	Mag	
0	10	5.41 ± 0.16			
5	10	6.63 ± 0.49			
10	13	8.25 ± 0.33			
30	10	12.27 ± 0.74			
61	10	20.79 ± 1.75			
90	11	26.55 ± 2.22	5		
180	14	49.15 ± 4.04	5		
215	10	52.75 ± 4.28			
243	11	51.93 ± 2.04			
271	9	57.38 ± 2.86		9	

Figure 1 Morphological features of Delta Smelt and their otoliths.

(A) Larval (left) and adult (right) Delta Smelt and associated length measurements: standard length (SL), total length (TL), fork length (FL, adult only). (B) Ontogenetic development and morphology of sagittal otoliths from individuals of ages 30 to 271 days-post-hatch (dph). Dorsal-ventral (DV) and rostral-postrostral (RP) dimensions, and the dorsal aging trajectory (black/white line) are shown for the 180 dph otolith. (C) A polished sagittal section of the dorsal lobe of a Delta Smelt otolith exhibiting 180 daily rings from the outer edge of the core to the outer edge of the otolith (death). Tick marks in (C) represent 30-day intervals along the aging trajectory. Delta Smelt artwork by Adi Khen.

Otolith preparation

Sagittal otoliths from larval, juvenile, and adult Delta Smelt (Fig. 1A) were dissected and mounted using standard methods, adapted for the size of each age class (Hobbs et al., 2007; Hobbs et al., 2019). Otoliths from larvae were dissected using 30-gauge hypodermic needles and mounted on top of a drop of Loctite Super Glue and imaged with a drop of glycerin at 1000× magnification. Adults and juveniles were dissected using size 10 scalpel blades and ultra-fine tip forceps. Prior to sanding and polishing, whole, intact otoliths of fish ≥ 180 dph were imaged at 40× magnification, while those of fish between 30–90 dph were imaged at 200× magnification. All whole otolith images were taken with an Amscope MU1000 10-Megapixel camera on an Olympus CH30 compound microscope. The rostrum-postrostrum and dorsal-ventral measurements were digitally measured using ImageJ (version 1.8.0). After imaging, otoliths were mounted in the sagittal plane to glass microscope slides using Crystal Bond thermoplastic glue and stored in plastic microscope slide boxes.

Mounted otoliths were wet sanded with 600, 800, and 1200 grit Buehler MicroCut silicon carbide paper and polished with 0.3-µm Buehler MicroPolish alumina on a Buehler Microcloth (Buehler, Lake Bluff, Illinois, USA) on the sulcus side, then flipped and sanded to expose the core (the section of the otolith around the primordium, the origin of otolith material deposition, that is bound by the hatch mark) and daily increments. All polished otoliths were imaged at 200x, with additional images captured at 400× magnification for 271 dph fish (Table 1). All images were taken using an Amscope MU1000 10MP camera on an Olympus CH30 compound microscope and stitched together using the photo merge function in Adobe Photoshop 2020 (v. 21.1.1). Left otoliths were initially sanded; however, if the left otolith was broken, lost, or of poor quality, the right otolith was prepared in its place. For the comparison of otolith symmetry, a subset of fish had both the left and right otoliths prepared: 90 dph (n = 5) and 180 dph (n = 5) age groups (Table 1). In total, 108 Delta Smelt (120 otoliths) were examined achieving approximately 10 samples in each age class (Table 1) for individuals aged 0–271 dph and 5.41 to 57.38 mm SL (Table 1).

Otolith age and growth analyses

The quality of each otolith image was ranked on a scale of 0 to 3 (low to high, respectively) based on the clarity of the core and edge increments, with only quality 2 and 3 otoliths used in analyses. All otoliths were analyzed with ImageJ by three independent analysts without prior knowledge of each fish’s age. Images were calibrated from pixels to µm using a stage micrometer. Increments were counted from the core to the dorsal edge, excluding embryonic rings present within the core, which provided the most consistent and clear age trajectory on the otolith (Hobbs et al., 2007) (Fig. 1B). A distinct hatch mark, indicating the moment larvae emerge from eggs into the ambient environment, was observable in each otolith as a thick, dark band approximately 8 µm from the primordium. This was used to identify the first daily increment from which the growth profile was constructed for each otolith (Fig. 1C, “core”).

Accuracy

Accuracy was quantified to assess how well otolith-based age estimates reflect the known ages of cultured Delta Smelt. Error in accuracy (EAfi) of a given age estimate for a given fish, reflecting both accuracy and bias (in days), was calculated as the raw deviation from the known age of the fish: (1) EAfi=afi−af ^

where afi is the i th age estimate and af ^ is the known age of the f th fish. Percent error in accuracy (PEAfi) of a given age estimate, an age-normalized estimate of the absolute error, was calculated as 100 times the ratio of the absolute error and the known age of the f th fish. (2) PEAfi=|EAfi|af ^ ∗100

Precision

Precision was quantified to assess the reproducibility of repeated age estimates among the three independent analysts (Campana, 2001). Error in precision (EPfi) of a given age estimate for a given fish was calculated as the raw deviation (in days) from the mean age estimate of the fish: (3) EPfi=afi−af ¯

where af ¯ is the mean age estimate for the f th fish. Percent error in precision (PEPfi) for a given fish, an age-normalized estimate of the absolute inter-operator error, was calculated as 100 times the ratio of the absolute precision error and mean age estimate of a given fish: (4) PEPfi=|EPfi|af ¯ ∗100

Statistical analyses

First, to describe the general somatic growth curve for cultured Delta Smelt, size-at-age was modeled using a Gompertz growth function with the known ages (t = time in, dph) of cultured fish and their standard lengths (SLc), where kg is the growth rate coefficient, A is the upper asymptote, and c is related to the time at inflection (Tjørve & Tjørve, 2017) (Eq. 5). (5) SLc∼A∗e−c∗e−kg∗t

Fish size was then contrasted with otolith size using simple linear regression to assess otolith-size to fish-size (OS-FS) proportionality, and OS-FS proportionality was contrasted among cultured and wild adult Delta Smelt using a linear model. To examine the additive and interactive contributions of analyst identity, otolith side, and fish age on the accuracy (PEA) of otolith-based age estimates, a subset of larval (90 dph, n = 5) and adult (180 dph, n = 5) Delta Smelt were selected to have both the left and right otoliths analyzed. Both otoliths from each fish were aged by each of the three analysts, and a linear model was constructed to examine the additive and interactive effects of analyst identity (I), otolith side (O), and age class (A) on the accuracy (PEA) of age estimates (Eq. 6). (6) PEA=β0+β1I+β2O+β3A+β4IO+β5IA+β6OA+β7IOA+ε

ε∼N0,σ2

To examine the interactive effects of microscope magnification and analyst identity on the accuracy (PEA) of otolith-based age estimates for the oldest, slowest growing individuals (with the smallest rings), one otolith for each of the age-271 dph Delta Smelt (N = 9) was imaged at both 200x and 400x magnification, with both images being aged by each of the three analysts, and a linear model was constructed to examine the additive and interactive effects of analyst identity (I) and image magnification (M) on the accuracy (PEA) of age estimates (Eq. 7). (7) PEA=β0+β1I+β2M+β3IM+ε

ε∼N0,σ2

All ordination and modeling were conducted in the R software environment version 3.6.3 (R Core Team, 2019). The Gompertz model was fit using the “nls” function, which determines the nonlinear (weighted) least-squares estimates of the parameters of a nonlinear model. Models were run and compared using maximum likelihood estimation while assuming a Gaussian distribution. Model assumptions were examined using Q-Q and residual plots. Likelihood ratio tests were used to assess the significance of each model relative to the null model (intercept only) with α = 0.05.

Results

Somatic and otolith growth

The Gompertz growth model provided a reasonable fit to size-at-age of cultured Delta Smelt (A = 62.15, c = 2.20, kg = 0.012, R2 = 0.994) (Fig. 2A). Overall, otolith size varied strongly and linearly with fish size (slope = 0.070 ± 0.001, intercept = 6.79, R2 = 0.988), indicating that otolith growth is largely proportional to fish growth across the age classes examined (Fig. 2B). The smallest larvae, however, exhibited an inflection at approximately 10 mm SL (otolith radius = 18.96 µm), with newly-hatched larvae <10 mm (∼20 dph) exhibiting a steeper slope of 0.282 ± 0.03, and all older individuals exhibiting a constant slope of 0.069 ± 0.001, similar to the global slope of 0.070 (Fig. S2, Table S1). Mean ± s.d. otolith-somatic ratios of juvenile-adult (>35 mm) cultured and wild Delta Smelt were 12.0 ± 0.8 and 12.27 ± 1.0 µm/mm, respectively, and did not differ significantly between the two groups (t = −1.82, df = 97.3, p = 0.072, Fig. 2B). Otoliths exhibited a single primordium with core sizes (otolith radius at hatch) of 8.5 ± 1.5 µm (mean ±s.d.) (Fig. 2C), corresponding with a size-at-hatch of 5.41 ± 0.17 mm (mean ± s.d.) standard length. Otolith growth profiles exhibited ontogenetic variation in accretion rates, with slower rates of 1–2 µm/d for fish <30 dph, increasing to over 8 µm/d in 50–100 dph fish, followed again by a gradual decrease back to 2 µm/d as fish matured toward 270 dph (Fig. 2D).

Figure 2 Somatic and otolith size and growth in Delta Smelt.

(A) Standard length versus age and fitted Gompertz growth curve, (B) otolith size (radius) versus standard length and fitted linear model, (C) density plot of otolith core sizes, and (D) otolith growth trajectories with known ages (vertical dashed lines) examined in the present study. In panel (B), “X” symbols represent wild Delta Smelt captured in 2019 and the inset boxplot shows the otolith-somatic size relationship for wild and cultured fish >35 mm.

Accuracy and precision

Error in accuracy changed from 0 to 10 days as known ages increased from 5 to 271 dph, indicating negative bias in older specimens (Fig. 3A). Absolute percent error in accuracy declined from 20% to 2% in fishes of age 5–90 dph, increasing to 3–4% in older 243–271 dph fish (Fig. 3B). Due to the sensitivity of percent error to small deviations in fish <10 dph, mean error and percent error in accuracy and precision were only contrasted for fish ≥ 10 dph. Mean EA and PEA across all age classes (10–271 dph) and the three analysts were −3.6 days and 3.9%, respectively, and values for each were similar among analysts (Figs. 4A–4B). Mean percent error in precision across all age classes and analysts was 2.2% (Fig. 4C), and also was similar among analysts. These errors correspond with relatively high accuracy (96.1%) and precision (97.8%), thus confirming the proper identification of the hatch mark and first increment, daily periodicity of increment formation, and both accurate and repeatable age estimation by multiple independent analysts.

Figure 3 Aging accuracy across age (days-post-hatch).

Mean ± SD error (A) and percent error (B) in the accuracy of age estimates for Delta Smelt are plotted in relation to the known age for each group.

Figure 4 Aging accuracy and precision among analysts.

(A) Raw error in accuracy (EA in dph), (B) absolute percent error in accuracy (PEA, %), (C) raw error in precision (EP in dph), (D) absolute percent error in precision (PEP, %). Points and bars represent the mean error value for each analyst; segments represent 1 s.d. The global mean for all analysts is shown by the red dashed line. Only specimens ≥ 10 dph were included in the analysis. Statistical results in Table 2.

Effects of otolith symmetry, image magnification, and life stage on aging accuracy among analysts

The 90-dph and 180-dph fish, for which both otoliths were analyzed, exhibited mean absolute percent error in accuracy (PEA) of 1–2%, which did not appear to vary as additive or interactive functions of life stage, otolith side, or analyst (Table 2, Fig. 5). For the oldest fishes examined (271 dph), however, error was often biased 7–10 days lower than the known age (Fig. 4A), suggesting that the smaller increments in these slower-growing specimens were often inconspicuous and overlooked. Re-imaging of older specimens at higher magnification (400x versus 200x) yielded significant improvement in age estimates for older fish, reducing mean bias to 0.25% (Table 3, Fig. 5).

Figure 5 Variation in aging accuracy in relation to otolith side and image magnification.

Raw error (EA, dph) among left versus right otoliths (A) and 200× versus 400× images (B). Absolute percent error (PEA, %) among left versus right otoliths (C) and 200× versus 400× images (D). Comparisons between left and right otoliths included fish ages 90 and 180 dph, whereas comparisons among magnifications focused on the slowest-growing 271-dph age class. Statistical results in Table 3.

Discussion

Experimental validation of otolith-based metrics

Validation studies are essential for assessing confidence in otolith-derived metrics (Campana, 1990; Campana, 2001). Our results build upon prior work to refine and expand our understanding of the accuracy and precision of otolith-based tools across multiple life stages of a critically endangered estuarine fish. Using cultured, known-age specimens and multiple independent analysts, we demonstrated daily periodicity of increment formation and high accuracy and precision of age estimates. The limited asymmetry observed between left and right otoliths suggested that either otolith can likely be utilized, and increased magnification improved aging precision, indicating that otolith approaches can be applied to older fish with slower accretion rates. In aggregate, our results indicate that the application of otolith-based techniques to archived collections of Delta Smelt can yield repeatable, accurate, and valuable estimates of the hatch dates, age structure, growth rates, and timing of movements (when paired with otolith chemistry) across life stages (Hobbs et al., 2007; Hobbs et al., 2019). Such experimental approaches using multiple independent analysts and known-age specimens across key developmental stages can greatly improve confidence in otolith-derived metrics that are valuable for informing resource management and species conservation.

Table 2 Effects of observer, otolith side, and life stage on aging accuracy.

Results of a linear model examining the effects of observer (obs), otolith side (left vs. right), and fish age group (90 vs 180 dph) on the absolute percent error in accuracy (PEA) of age estimates. Factors include: obs-analyst, oto-otolith side, known-known age. No significant effects were detected (P > 0.05).

Factor	DF	SS	MS	F	P	
obs	2	16.49	8.25	2.44	0.098	
side	1	3.22	3.22	0.95	0.335	
age	1	0.25	0.25	0.07	0.786	
obs:side	2	6.06	3.03	0.90	0.415	
obs:age	2	1.31	0.65	0.19	0.825	
side:age	1	1.16	1.16	0.34	0.561	
obs:side:age	2	2.32	1.16	0.34	0.712	
Residuals	48	162.47	3.38			
Notes.

DF degrees of freedom

SS sum-of-squares

MS mean squares

Table 3 Effects of observer and image magnification on aging accuracy.

Statistical results of the linear model examining the additive and interactive effects of observer (obs) and image magnification (mag: 200× vs 400×) on the absolute percent error in accuracy (PEA) of age estimates for the slowest-growing 271 dph age group. Significant P values are in bold.

Factor	DF	SS	MS	F	P	
obs	2	15.95	7.97	2.98	0.059	
mag	1	86.29	86.29	32.30	<0.001	
obs:mag	2	36.07	18.04	6.75	0.002	
Residuals	54	144.27	2.67			
Notes.

DF degrees of freedom

SS sum-of-squares

MS mean squares

Interpretation of otolith microstructures

For many fish species, the first otolith increment is visible following a dark, thick “hatch check” which is assumed to form when larvae emerge from eggs into the ambient environment (Campana & Neilson, 1985; Ohama, 1990). However, verification of the timing of such features is necessary to facilitate accurate age reconstructions for each species. For example, features identified as hatch checks may correspond with other processes, such as the end of the yolk-sac stage or first-feeding (Moksness, 1992; Sepulveda, 1994; Hirose & Kawaguchi, 2001), and embryonic rings can form prior to hatch and with unknown periodicity (Stevenson & Campana, 1992). In Delta Smelt otoliths, a distinct hatch check was consistently observed at approximately 8–9 µm from the center of the otolith primordium. We also observed embryonic rings of unknown periodicity that are generally excluded from increment analyses (Stevenson & Campana, 1992). Following the hatch check, increments were accreted daily as evidenced by strong agreement between increment counts and known ages of each fish. Though the first observations of Delta Smelt otoliths suggested that increments first formed at 5 dph (Hobbs et al., 2007), we believe this was likely due to the lower resolution (e.g., 1 megapixel) digital cameras available at that time, with lower image quality obscuring the smallest larval increments. Thus, Delta Smelt exhibit clear microstructures including a single primordium, a prominent hatch check, and daily increments that, together, facilitate accurate reconstructions of hatch dates and size-at-age.

Otolith-somatic proportionality

A simple linear model indicated that 99% of the variation in fish size (SL) could be explained by otolith size across the range of age classes examined (0–271 dph). This result indicated that, for Delta Smelt, otolith-somatic proportionality is largely constant—a key assumption of standard models used to estimate size and growth rates from otolith radii and accretion rates, respectively (Campana, 1990). Furthermore, otolith-somatic ratios differed by <3% between cultured and wild fish, suggesting that otolith-based tools likely can be applied to the wild population (Fig. 2B). Closer inspection of otolith-somatic proportionality immediately after hatch (e.g., fish <10 mm or 20 dph), indicated that prolarvae may exhibit rapid somatic growth immediately after hatching despite low rates of otolith accretion; a phenomenon also observed in early otolith work on Delta Smelt larvae (Hobbs et al., 2007). During this early larval period, elongation (growth) was proportional to otolith growth, but exhibited a higher slope than the value observed in larger (>10 mm) fish (Fig. S2). This early larval stage is heavily reliant on endogenous sources of nutrition such as the yolk and oil globule (Mager et al., 2004). At approximately 10 dph, endogenous reserves are exhausted and larvae become dependent on exogenous feeding, and at 20 dph (10 mm), fin differentiation is initiated and at 20 dph (10 mm), fin differentiation is initiated (Mager et al., 2004). It is at this time when the somatic-otolith relationship decreased abruptly and remained constant thereafter for all subsequent age classes examined. This short-lived discontinuity in slopes had little influence on the overall somatic-otolith relationship (m = 0.069 vs 0.072, Fig. S2) for estimating the growth of older life stages, but would be important for otolith-based growth and size estimates for the youngest larvae (Fig. S2). In sum, otoliths of Delta Smelt exhibited constant proportionality for life stages > 10 mm with results similar among cultured and wild fish, thus further verifying their value for estimating somatic growth rates (Campana, 2001; Hobbs et al., 2007).

Accuracy and precision of otolith-based age estimates

Validation experiments can be relatively challenging for sensitive, critically endangered species such as Delta Smelt. Here, we leveraged ongoing efforts by the UC Davis Fish Conservation and Culture Laboratory to maintain a captive population of Delta Smelt for both conservation and research purposes, which provided a unique opportunity to build an archive of otoliths from known-age individuals across multiple age classes. Such an archive is uniquely valuable for assessing the accuracy and repeatability of otolith-based size, age, and schlerochronological reconstructions. Using known-age cultured fish and multiple independent analysts, we demonstrated relatively high (>95%) accuracy and precision of otolith-based age estimates for Delta Smelt.

Obtaining acceptable levels of accuracy and precision in age estimates from otoliths requires finely calibrated otolith preparation and imaging protocols, and subsequent training of analysts, to facilitate clear and accurate interpretations of daily increments. With careful preparation, daily increments can be observed in otoliths of Delta Smelt as fine rings (microstructures) that exhibit a consistent and readily-interpretable appearance, as observed in many fishes (Stevenson & Campana, 1992). Poor preparation, however, can result in multi-day check marks or smaller sub-daily rings that generate systematic bias in age and growth reconstructions (Campana & Neilson, 1985; Stevenson & Campana, 1992). For example, under-sanding can result in thick samples that emphasize larger multi-day checks that obscure the true daily increments, while over-sanding can result in extra thin samples that emphasize sub-daily otolith features or image artifacts that occur at a higher frequency than the true daily increments. For these reasons, archives of known-age specimens are needed, proper quality assurance and control (QAQC) procedures should be employed, and only high-quality preparations and well-trained analysts should be used in otolith-based studies of wild populations, where precision is assessed while accuracy is assumed (Campana, 1990; Hobbs et al., 2019).

Otolith symmetry

Paired otoliths in fishes may exhibit varying degrees of asymmetry (Lychakov et al., 2006; Lychakov et al., 2008; Díaz-Gil et al., 2015). Symmetrical otoliths can be valuable for studying rare species, where each specimen is highly valuable. For example, to control for potential effects of asymmetry (Mahé et al., 2019), otolith protocols often use only left or right otoliths, replacing entire specimens if the selected otolith is lost, damaged, or otherwise unusable for growth analysis. For valuable specimens of endangered species, however, each specimen is difficult to replace, thus the use of either otolith (e.g., if the preferred otolith is lost or damaged) is often valuable, when shown to be appropriate (Hobbs et al., 2007; Hobbs et al., 2019). If left and right otoliths exhibit systematic asymmetrical accretion patterns, however, this can greatly affect otolith-based inferences when otolith side cannot be standardized. Here, we showed that age estimates from paired left and right Delta Smelt otoliths were similar, with no evidence for differences in accuracy or precision. These results suggest that reliable age and growth estimates can likely be constructed from either left or right otoliths of Delta Smelt.

Image magnification

During QAQC of image quality, older Delta Smelt specimens (e.g., 271 dph) often exhibited increment compaction toward the otolith edge, indicative of the ontogenetic reduction of growth rate in older fish. These rings were occasionally flagged as “low quality” by analysts due to their poor appearance when imaged at 200× magnification. Though these samples exhibited acceptable percent precision and accuracy (∼95%), the increase in the raw daily error rate and consistent negative aging bias for these older individuals indicated that the finest daily rings were often missed by each analyst for these slow-growing specimens. By re-imaging the otoliths from the oldest 271 dph fish at twice the magnification (400×) and re-analyzing the microstructures, the accuracy of age estimates increased significantly and exhibited no measurable bias. Though higher-resolution imaging requires additional time and computing power for capturing, storing, and processing larger datafiles, our results indicate that this additional effort significantly improved otolith-based age and growth estimates for older age classes of Delta Smelt.

Cultured versus wild fish

The limitations of studying wild Delta Smelt include the species’ sensitivity to handling and its rarity and protected status as a critically endangered species. The use of cultured fish was valuable for providing known-age specimens to verify the interpretation of otolith microstructures. The development and interpretation of otoliths in wild fish, however, could be more complex than for those in controlled laboratory studies (Campana, 2001). For example, environmental variability may influence otolith shape, increment appearance, or the timing of the first increment (Campana & Casselman, 1993; Otterlei, 2002; Vignon & Morat, 2010). Here, we demonstrated that otolith-somatic proportionality was similar among cultured and wild sub-adult Delta Smelt. Furthermore, standard back-calculation tools such as the Biological Intercept Model explicitly account for individual variation in otolith shape by proportionally adjusting growth estimates using individual-based somatic-otolith relationships (Campana, 1990). Nevertheless, further comparisons between cultured and wild fish could be valuable for assessing confidence in otolith-based inferences for the wild population. If possible, additional field-based studies using caged or marked-recaptured specimens could provide further insights regarding the interpretation of otoliths from wild Delta Smelt.

Conclusions

With Delta Smelt rapidly approaching extinction in the wild, it has become necessary to address increasingly complex questions for a wider variety of life stages with relatively few wild specimens. Otolith-based studies, once validated across all relevant life stages, can provide valuable estimates of the size, age, growth, and phenology of individual fish to inform population models and directed conservation efforts. Here, we experimentally evaluated the accuracy and repeatability of otolith-based tools across multiple life stages of a critically endangered estuarine fish. Results indicated relatively high aging precision and accuracy, as well as similarity among cultured and wild individuals in otolith-somatic relationships. This study expands the validated otolith age range for Delta Smelt by 300% while providing an improved understanding of how key methodological factors are likely to affect confidence in otolith-based results. Such experimental assessments of the accuracy and repeatability of schlerochronological reconstructions remain key to their effective use in fisheries management and conservation.

Supplemental Information

Supplemental Information 1 Length conversion functions

Length conversion equations for larval and adult Delta Smelt measured fresh by hand or digitally using image analysis (A–B), preserved in ethanol (C–D) or frozen at −20 °C (E–F), and fork length (FL) conversions from SL and TL of frozen specimens. Dashed lines represent 1:1; red lines represent the respective linear models. Linear models and R2 values are proved in the top-left of each plot; proportional adjustment functions are provided in the bottom-right (C–H).

Click here for additional data file.

Supplemental Information 2 Examination of stage-specific somatic-otolith size relationships

Plots showing all size classes (A) and fish < 25 mm (B) are provided. Three models were fit including a global model (all size classes as in Fig. 1B, black), a model for pro-larval fish ≤ 10 mm (blue), and a model for all fish > 10 mm (red). Slopes (m) of each model are provided in (b) (see Table S3 for model details).

Click here for additional data file.

Supplemental Information 3 Results of stage-specific linear models examining somatic-otolith size relationships

Table headings include: Model = fish size class, Coefficient = slope or intercept for each linear model, Estimate = coefficient value, Error = standard error, t = t-value, P1 = p-value for each coefficient, n = sample size, F = F-statistic, P2 = p-value and R2 = coefficient of determination for the model. SL = standard length.

Click here for additional data file.

Supplemental Information 4 Merged fish and otolith data

Raw data on fish size, otolith size, and individual age transects are provided.

Click here for additional data file.

We thank the many staff of the UC Davis Fish Conservation and Culture Laboratory for their assistance in rearing and archiving known-age Delta Smelt for this study. The UC Davis Department of Wildlife, Fish, and Conservation Biology (N. Fangue) and Center for Aquatic Biology and Aquaculture (L. Deanovic) provided laboratory space and logistical support. Constructive reviews from N. Bertrand and several anonymous reviewers greatly improved the manuscript. The views expressed herein are those of the authors and do not represent the official opinion of any employer, institution, or government agency.

Additional Information and Declarations

Competing Interests

Author Contributions

Animal Ethics

Data Availability

The authors declare there are no competing interests.

Wilson Xieu and Levi S. Lewis conceived and designed the experiments, performed the experiments, analyzed the data, prepared figures and/or tables, authored or reviewed drafts of the paper, and approved the final draft.

Feng Zhao and Rachel A. Fichman conceived and designed the experiments, performed the experiments, analyzed the data, authored or reviewed drafts of the paper, and approved the final draft.

Malte Willmes analyzed the data, prepared figures and/or tables, authored or reviewed drafts of the paper, and approved the final draft.

Tien-Chieh Hung, Luke Ellison, Troy Stevenson and Galen Tigan conceived and designed the experiments, performed the experiments, authored or reviewed drafts of the paper, and approved the final draft.

Andrew Schultz conceived and designed the experiments, authored or reviewed drafts of the paper, and approved the final draft.

James Hobbs analyzed the data, authored or reviewed drafts of the paper, and approved the final draft.

The following information was supplied relating to ethical approvals (i.e., approving body and any reference numbers):

Institutional Animal Care and Use Committee of the University of California, Davis provided approval of this research (IACUC Protocol (19747).

The following information was supplied regarding data availability:

The raw data are available in the Supplemental File.

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
