# Peer review of "Experimental validation of otolith-based age and growth reconstructions across multiple life stages of a critically endangered estuarine fish"

_PeerJ, doi:10.7717/peerj.12280_

## Round 0.1 · original submission · Major Revisions

Thank you for submitting your manuscript to PeerJ. The two reviewers of your paper were positive about its contributions but noted some concerns and made suggestions that I invite you to address. Reviewer 1 suggests edits to the introduction and discussion and recommends that you set your results in a broader context. Reviewer 2 writes that the extrapolation of your results to wild-populations is not supported by the current work and suggested solutions. They also suggested that terms be defined more thoroughly. Lastly, reviewer 2 had questions about whether the paper is a self-contained study or single “unit of publication” as defined by PeerJ.

I invite you to submit a revised manuscript and ask that your rebuttal letter address all reviewer comments.

I look forward to receiving your revised submission.

Reviewer 1 ·

Basic reporting

The terms used for naming of the sectioned otolith internal morphology vary among fish species and authors. The term “core” used by the authors here would be acceptable if precisely described and used consistently in the whole text. Nevertheless, I am afraid that it is not. For the first time the term “core” appears in the 153 to express the inner central part of the otolith without a precise definition what it means. In line 164 the phrase “based on the clarity of the core and edge increments” may suggest that there are increments deposited in the “core” and they are important for further analyses. In line 167 there is a citation: “Increments were counted from the core to the dorsal edge… (Hobbs et al., 2007) (Fig. 1B).” which could be misleading as it is not clear if increments deposited in the core were counted only by Hobbs et al., (2007) or also by the authors (Fig. 1B). From the next sentence (lines 168-170) I got an impression that the term “core” has been synonymized with “primordium” (a center of otolith deposition) as a distance between the core and hatch mark was given (8 μm). In the following sentence (lines 170-172) it is clarified where the daily increments enumeration starts, i.e. from a hatch mark towards otolith dorsal margin which could imply that the “core” was used to describe the most inner part of the otolith section located around the single central primordium and externally limited by a hatch mark. Although the reference to Fig. 1c (Fig. 1C, “core”) at the end of this sentence introduced information noise again. The caption of Fig. 1C says: “(C) … 180 daily rings from the core (hatch) to the outer edge (death)”. Moreover, descriptions and symbols in Fig. 1C explain that the most inner part otolith section (arrow) name is “primordia” (should be “primordium” as Delta Smelt otolith is characterized by a single central primordium), while a “core” indicated by a vertical line refers to the hatch mark! It is not a big deal to make some order with these terms. I would suggest sticking to the following: “central primordium” as a origin of otolith growth deposition and starting point of the radius measurements and a “core” - otolith section located around the single central primordium and externally limited by a hatch mark.
The figure 3 description: “Aging accuracy across life stages.” does not match the figure contents, as the aging accuracy was presented vs. age and life stages were not indicated on the X axis.
According to the instructions for reviewers listed under the final subtopic of “Basic reporting:”, i.e. Self-contained with relevant results to hypotheses. (The submission should be ‘self-contained,’ should represent an appropriate ‘unit of publication’, and should include all results relevant to the hypothesis. Coherent bodies of work should not be inappropriately subdivided merely to increase publication count.) I have some doubts that the submitted work constitutes a part of a larger project thus representing a “salami slicing” case. Of course the reviewer cannot prove such a serious accusation but if one considers the arguments listed below and in the next paragraphs of the review (Experimental Design and Validity of the Findings), such a possibility should be considered. A validation of a daily character of growth increments in otoliths should constitute the first step towards facilitation of age and growth studies of fast growing/short life-span fish or early life history stages. However, opposite to the reviewed paper, in most publications such a protocol is a part of a larger project. In the reviewed paper the participation of 11 coauthors from six scientific and research institutions appeared to be necessary to verify the already published findings of Hobbs et al. (2007) and to extend the age-range of the studied fish from about 100 (Hobbs et al. 2007) up to 271 days post hatch. Both studies were performed on the cultured material. Therefore, the extension of the otolith microstructure by the reviewed paper on larger and older fish does not give the final answer if the wild fish otoliths of adults can be aged on the basis of daily growth increments.

Experimental design

The research questions/targets defined in lines 96-103 and the outcome of the performed analysis presented throughout the paper from the abstract to conclusions make my ambivalent expressions. From a positive perspective everything is well defined, processed, methods are described with sufficient detail, statistically sound and relevant. However, if we consider the whole performed research is scientifically justified, some doubts arise. The executed research was based on the cultured fish kept in stable environmental conditions and feeding regime relevant for consecutive live history stages. The lack of reference to a wild fish otolith microstructure and growth greatly undermine the importance of the proposed research. Thus, I am afraid that the presented conclusions are reaching too far. Authors, on the basis of earlier published papers are aware that results obtained with cultured fish material should not be extrapolated directly on wild fish population, especially those with a complex migratory behavior, where the impact of environmental conditions may greatly disrupt otolith growth - somatic growth relationship, growth increments width or their definition. However, to support their conclusions the authors mentioned that “prior work has indicated that such variation is likely negligible for reconstructing size-at-age in Delta Smelt ≤ 90 mm SL (Hobbs et al.,2007)”. I have checked the Hobbs at al. (2007) as carefully as the short time for this review allowed but I could not find the mentioned phrase. Therefore, the extension of the observations achieved from the experiments and materials based on cultured fish on to wild populations, in my opinion is not fully justified (lines 407-410). However, all the presented results, if limited to the cultured fish, are achieved according to the highest standards of otolith and fish growth research. Especially those on the otolith-somatic proportionality and initial increment formation. Let me also express my doubts if otolith microstructure is utilized in the future to study cultured Delta Smelt growth as there are much simple and cost effective methods to estimate growth when the hatch date is known. In addition, I am sure that the effects of image magnification for older, slower growing individuals has not been studied so far and possibly will not be studied that way by anybody else in the future.

Validity of the findings

As already mentioned under the previous paragraph of the review, not every conclusion is supported by the results and fully justified. In addition to that in the discussion (lines 347-351) there are two sentences not supported by the submitted paper results thus should be removed from the manuscript or a relevant citation added.

Additional comments

In addition to several technical comments regarding terminology and figures or figure captions, I have the following suggestion how to make the carefully elaborated results based on culture fish publishable. It is possible on a low labor input and project cost but additional materials should be available. The authors need to convince the audience that the achieved results and “experimental approaches greatly improve the interpretation of otolith-derived metrics for fisheries ecology and conservation, the facilitating valuable reconstructions of hatch dates, growth rates, and movement patterns of wild fish populations”. And without a reference to the wild Delta Smelt population/s it does not seem to be possible. In addition, to the already performed research the relationship otolith size vs fish length of cultured fish and wild fish should be compared. This relation could act as a proxy of the somatic growth rates according to the well-known rule that faster growing fish (younger) deposit relatively smaller otoliths than slow growing (older) fish of the same length. If there is no significant differences between those two groups your observations are easy transferable to the wild population. Remember to use the otolith length rather that otolith radius and for adults both sexes should be treated separately to avoid additional noise if there is a dimorphic growth in this species. Otolith length – fish length relationship could be also utilized to calculate the size structure of Delta Smelt from predator’s stomachs and/or for the assessment of the species size structure from otoliths found in bottom deposits. Of course such a simple approach does not exclude a need of more specific studies of otoliths from wild populations.

Reviewer 2 ·

Basic reporting

Fails
Literature references, sufficient field background/context provided (see below and suggestions in marked MS)
1 the introduction doesn’t provide enough background (i.e. is there other known age experiments)
2 the introduction could benefit from a restructure. At the moment we hear about the study spp in the first parag. This would be best left to late in the intro. Instead start broadly ie brief on mgt of threatened fish or biota, hard parts to gather demographic info, background on otoliths, then the study spp, objectives….
3 the intro doesn’t cite the literature beyond campana and a couple of target spp studies
4 the discussion needs to put the study in a wider context (e.g. schlero is first mentioned in the conclusion). How bout some references to blacks work or gillanders. Ie the exciting aspect is this is all validated now we can compare growth across seasons and different habitats
5 What does the different increment width at ~ 60 d mean - expand
6 the concern with applying this information is that real world noise may make the microstructure different. As such, there needs to be a discussion about how a marked wild fish trial is an important next step and seek to find if that has been done before and put the study in that context.

Experimental design

Pass (aims and scope of journal not clear)

Validity of the findings

Fails
impact and novelty not assessed. Negative/inconclusive results accepted. Meaningful replication encouraged where rationale & benefit to literature is clearly stated. – expand context
Conclusions are well stated, linked to original research question & limited to supporting results. – mentioned concern on conclusion above

Additional comments

Congratulations on a well designed study that is important for conservation of threatened fishes. However, before publishing the intro and discussion need work.

---

## Round 0.2 · Minor Revisions

Thank you for submitting your revised manuscript to PeerJ. The reviewers are positive about your revisions and have some minor suggestions for you to consider. Reviewer 1 has a question about the linear regression models used to relate otolith and fish size, and whether this relationship is constant across all age classes. They also express a concern about the statistic used to compare otolith/somatic ratios between cultured and wild delta smelt. Reviewer 2 lists a few minor suggestions to consider.

I invite you to submit a revised manuscript after minor revisions and ask that your rebuttal letter address all reviewer comments.

I look forward to receiving your revised submission.

Reviewer 1 ·

Basic reporting

As it is my second revision of the second draft of the same paper I have inserted all mu comments under the p.4 "Additional comments"

Experimental design

As it is my second revision of the second draft of the same paper I have inserted all mu comments under the p.4 "Additional comments"

Validity of the findings

As it is my second revision of the second draft of the same paper I have inserted all mu comments under the p.4 "Additional comments"

Additional comments

I feel satisfied with an extensive scale of positive and merit content of the authors’ response on my comments, questions and critical remarks. Most of my reservations have been accepted or reasonably explained.

Despite of the above mentioned, positive authors’ response on the reviewers comments I have to express two additional doubts/remarks that should be eventually addressed by the authors during the second round of the reviewing process. To be honest I have overlooked one of these issues in my comments to the first draft but even so I consider it important enough to be mentioned at this stage.
246-248 Fish size was then contrasted with otolith size using simple linear regression to assess otolith-size to fish-size (OS-FS) proportionality, and OS-FS proportionality was contrasted among cultured and wild adult Delta Smelt using a linear model.
During reviewing the first draft, I have somehow omitted analysis of the Figure_S2 and Table S1 thus, the methodology described in lines 246-248 followed by statistics and the previous version of Figure 2B looked fine to me. Now, I am aware that the presented linear regression to describe otolith-size fish-size relationship, fitted to data representing the whole size range may result in certain limitations of the proposed model. Despite of a high r2 the single linear model for the whole size range of investigated fish will be not the best tool, e. g. for back-calculation of fish somatic growth from otolith increments. The residuals have not been analyzed and the noticeably steeper slope of the OS-FS relation during early larval period suggests that residuals are not distributed evenly.

Authors expressed some doubts towards a single general model for all size classes as mentioned in the Results and Discussion and in Table S2 where three options of linear models have been presented. I do agree that slight differences of coefficients for all sizes and >10 mm SL models could be practically neglected but due to a significant difference of the otolith size vs SL relationship in early larvae the sentence “Overall, otolith size varied strongly and linearly with fish size (slope = 0.070 ± 0.001, intercept = 6.79, R2 = 0.988), indicating that otolith growth is largely proportional to fish growth across the age classes examined (Fig. 2B)” is not fully justified. I do not understand why the authors have not calculated this relationship using e. g. stepwise (piecewise) linear regression as performed in many other otolith growth vs somatic growth studies? Changes of otolith size – fish size relationship at certain ontogenetic stage transitions (uncoupling of otolith growth and somatic growth) are common in many fish species. With regard to the assumption of constant proportionality between otolith growth and fish growth, the overall validity of constant proportionality becomes suspect when applied to the early life history stages of fishes (Hare and Cowen 1995).

My second reservation/concern refers to the sentence pasted below and the Fig. 2B.
283-285 Mean ± s.d. otolith-somatic ratios of juvenile-adult (> 35 mm) cultured and wild Delta Smelt were 12.0 ± 0.8 and 12.27 ± 1.0 μm/mm, respectively, and did not differ significantly between the two groups (t = -1.82, df = 97.3, p = 0.072, Fig 2B).
In my opinion the statistical procedures chosen to prove that otolith somatic ratios in individuals >35 mm of cultured and wild Delta Smelt do not differ are not appropriate. I do not want to discuss if p = 0.072 means that observed difference is significant or not. I would say that to compare if two linear regression functions differ or not the coefficients should be calculated for both and then tested. I cannot do it as the data for wild fish are not available. Moreover, comparison of otolith-somatic ratios on the basis of their mean values, even when performed to a similar range of fish size and statistically justified i.e. data fit to normal distribution, does not meet biological criteria of relative otolith size to fish body size growth. Even though otolith growth generally follows somatic growth trends, the relative otolith size to the fish size changes with fish size, age, life history traits, variations of environmental parameters and feeding conditions. Thus, a mean value does not constitute appropriate measure to investigate such a relationship between populations. I do insist to change the method to compare the otolith somatic ratios in individuals >35 mm of cultured and wild Delta Smelt from means to regressions coefficient analysis.
If may suggestions are accepted some slight changes should be introduced to the “Otolith-Somatic Proportionality” chapter of Discussion.

Reviewer 2 ·

Basic reporting

pass

very minor:
1 Concurrent sentences on lines 460 and 462 begin with the same ~ half dozen words. This is awkward and reducing the wordiness of the start of the sentence beginning in 462 will alleviate.
2 Sentence on ln 487 is back to front ie the first phrase and refs could be moved to the end and reduce clumsiness and readily make sense.
3 Species names in italics ie refs.

Experimental design

pass

Validity of the findings

pass

Additional comments

Much improved MS. Congrats to authors on a great contribution to fish, calcified structure and fisheries science.

---

## Round 0.3 · accepted · Accept

Thank you for your thorough consideration and response to the remaining reviewer comments. I am happy to now accept your paper for publication in PeerJ.

You will be given the option to make the reviews of your manuscript available to readers. Please consider doing so as this review record can be a great resource for readers of your paper and contributes to more transparent science.

Thank you for choosing PeerJ as a venue for publishing your work.